# Fabrication of CrSi_2_-Containing Master Alloys for Modification of Fe-Containing Intermetallic Compounds in Aluminum Alloys

**DOI:** 10.3390/ma15217836

**Published:** 2022-11-06

**Authors:** Jincheng Sun, Aiga Takahashi, Kennosuke Higashi, Takuya Yamamoto, Sergey Komarov

**Affiliations:** 1Graduate School of Environmental Studies, Tohoku University, 6-6-02 Aza Aoba, Aramaki, Aoba-ku, Sendai 980-8579, Japan; 2Department of Metallurgy, Materials Science and Materials Processing, Graduate School of Engineering, Tohoku University, 6-6-02 Aza-Aoba, Aramaki, Aoba-ku, Sendai 980-8579, Japan

**Keywords:** aluminum recycling, casting, iron intermetallics, powder metallurgy, CrSi_2_ particles

## Abstract

Aluminum contaminations, particularly iron, present a serious challenge to aluminum recycling technology. This is why many studies have focused on the reduction of detrimental effects of iron-containing contaminations through addition of elements such as manganese and chromium. However, the desirable modifying effect is often difficult to achieve because it would require concentrations of added elements greater than the allowable limits for many aluminum alloys. Thus, an alternative way to obtain the modifying effect, by using a much smaller amount of the modifying elements which are added to the aluminum melt as solid particles, was proposed in this study. The main goal of the present study was to investigate the possibilities of fabricating an aluminum master alloy by adding CrSi_2_ particles onto the surface of the vortex formed during mechanical agitation of molten aluminum. Two kinds of CrSi_2_ powder were used: one was commercial powder, and the other was self-synthesized CrSi_2_ via mechanical alloying by planetary ball milling. The results revealed that CrSi_2_ particles with a larger size penetrate the melt better. Particles of three kinds were found to exist in the Al melt after the addition of CrSi_2_ powder: (1) inclusions of eutectic origin formed at the last stage of crystallization, (2) mixtures of Al-Cr compounds and original CrSi_2_ particles and (3) original CrSi_2_ particles. Low melt temperatures and short treatment times were found to favor the fabrication of master alloys because they impeded the dissolution of CrSi_2_ particles into the Al melt and, thus, allowed one to fabricate the master alloy containing the particles of the second and third types.

## 1. Introduction

Al-Si-casting alloys are widely used for engineering applications, especially in the automotive industry, due to their high strength, lightweight and excellent castability. Faced with the increasing demands for energy saving, aluminum manufacturers are looking for more sustainable technological routes, particularly using more secondary aluminum for alloy production. However, this inevitably results in product quality loss due to contamination, especially by iron. That is why many efforts have been made recently to develop economically acceptable technology for the reduction of the detrimental effects of iron contamination. It is well known that the addition of some transition metals to Al-Si casting alloys has a beneficial effect on the morphology of Al-Fe-Si compounds. Particularly, the addition of elements such as manganese and chromium promotes the formation of α-AlFeSi compounds with Chinese script morphology. As a result, the formation of detrimental β-AlFeSi compounds can be suppressed, at least partly. This technique has been widely used over a variety of casting alloys, and its efficiency is especially apparent in high concentrations of iron when primary intermetallic compounds are formed. For example, Bjurenstedt et.al [1] used a commercial secondary aluminum alloy containing 1.19 wt.% of Fe to study the morphology and growth of primary α-Al(FeMnCr)Si intermetallics. Oda et.al [2] examined the ultrasonic refining effect using Al-Si-Fe-Mn alloys without or with the addition of chromium and titanium. The cooling-rate-dependent modification effect of Mn on the formation of the Fe-containing intermetallic phases of Al-Si-Mg secondary cast aluminum alloys was investigated by Cinkilic et.al [3]. Jin et.al [4] discussed the evolution of Fe-rich intermetallics in Al-Si-Cu 319 cast alloys with various Fe, Mo, and Mn contents, and the results showed that the better modification effect is achieved by the combined addition of Mo and Mn in the alloys with high Fe content. Although the underlying mechanisms are not fully understood yet, at least two mechanisms are responsible for the compound modification. Below is a brief outline of these mechanisms.

The first one is the classical heterogeneous nucleation on wettable particles, which should be available in the melt prior to AlFeSi compound formation. In addition to good wettability, the relative lattice misfit between substrates and the nucleation phase must not exceed a critical value which is, according to the Bramfitt approach [5], equal to 10%. A typical example of exploiting such a mechanism is the refinement of α-Al grains by adding an Al-Ti-B master alloy containing TiB_2_ particles. This mechanism is especially useful when the substrate substance is a thermodynamically stable compound, while the nucleation phase is an elementary substance or an alloy of simple composition, such as the above-mentioned TiB_2_ and Al, respectively. Intermetallic compounds, however, in many cases, have complicated chemical and phase compositions, varying depending on solidification conditions, particularly on the melt cooling rate. That is why it is extremely difficult, if not impossible, to discover an appropriate refiner and realize the above-mentioned nucleation mechanism for intermetallic compounds. Nevertheless, some similar nucleation-based approaches are being applied to modify some AlFeSi intermetallic compounds. The idea is that many transition elements can form binary and more complex intermetallic compounds in the aluminum melt and, thus, provide sites for the heterogeneous nucleation of AlFeSi compounds. However, the concentration of such transient elements in the alloys should be significant in order to achieve the desirable modifying effect. This is because the heterogeneous nuclei must be formed first at higher temperatures. For example, Dietrich et al. [6] reported that addition of Cr to an iron-containing AlSi_9_Cu_3_ alloy causes the formation of a less detrimental cubic α-Al-(Fe,Mn,Cr)-Si phase; however, this effect can only be achieved if the Cr concentration exceeds 0.5%. The reason was found to be the Al_13_Cr_4_Si_4_ phase which is formed earlier than the above-mentioned iron-containing phase and, thus, can serve as a nucleation site for this phase. In another work [2], the combined effect of Ti and Cr addition with ultrasound irradiation was shown to be responsible for the refinement and modification of Al-Fe-Si compounds, provided that the concentration of Ti and Cr is higher than 0.4%. The originating mechanism suggested was an earlier formation of Al-Si-Ti intermetallic compounds followed by the nucleation of Al-Cr-Si compounds on their surface. The introduction of ultrasonic vibrations in the melt during this stage resulted in the refinement of these compounds, which led to the formation of finer Al-Fe-Si compounds on their surface. In another group of research [7,8], it was reported that oxides, such as Al_2_O_3_, MgO or their spinel compounds, can serve as heterogeneous nucleation sites for some intermetallic compounds. However, no reasonable explanation was provided about the nucleation mechanism. It is obvious that no matter whether the heterogeneous nuclei are intermetallic or oxide phases, they have to satisfy the main requirement of the heterogeneous nucleation theory: to provide energetically favorable sites for the nucleation of new phases.

The second mechanism is associated with modification through altering the chemical composition of original compounds in such a way that their morphology changes in a desirable direction. A typical example of such a modification is the addition of Mn to the melt to transform needlelike harmful β-Al_5_FeSi phases to skeleton or flower-like α-Al(Fe,Mn)Si compounds. The optimal modification effect is achieved by replacing a part of Fe atoms with that of the addition element. As this approach does not require the formation of nucleus particles prior to the formation of AlFeSi compounds, it has a higher flexibility in the choice of addition elements and the variation of their concentration compared to the first mechanism. Nevertheless, the concentration is still to be kept at a significant level to obtain the desirable effect. For example, Zhang et al. [9] investigated the morphology of intermetallic compounds in A356 alloys containing Fe (1.0~2.5%) and Mn (0~1.625%). The results revealed that large-sized platelet-β-type Fe compounds can be modified into polyhedral or Chinese script shapes when the ratio of Mn/Fe exceeds 0.5, with the precise value being dependent on the cooling rate. Shabestari et al. [10] showed that Mn additions above 0.9% to A413 casting alloys with a Fe content of 2.5% refine the β-phase from large plates to a more numerous, compact and polyhedral form. Besides, additions of Sr in an amount of more than 0.1% were found to result in the change of the morphology of the intermetallics from plate to star-like.

Thus, both the above-mentioned approaches require significant additions of modifying elements to the Al-Fe-Si alloys to achieve the expected effects, and their concentration can go beyond the permissible limits for a given alloy. In the present study, we proposed an alternative way to obtain the modifying effect, with the use of a much smaller number of modifying elements which were added to the aluminum melt as solid particles. The particles are composed of elements, at least one of them being a transition metal having a modifying effect on the AlFeSi compounds. Also, these particles should have high thermal and chemical stability in the aluminum melt in order to avoid their rapid decomposition and/or dissolution. Particularly, in the present study, CrSi_2_ intermetallic particles were used as a source of chromium to change the morphology of AlFeSi compounds. Taking into account that any modifying element should be added into the melt as a master alloy, the main focus of the present study was to investigate the possibilities of fabricating an aluminum master alloy such as that containing CrSi_2_ particles. For this purpose, first, an attempt was made to synthesize and characterize CrSi_2_ intermetallic particles using the ball milling technique. Then, the synthesized particles were used in high temperature experiments to investigate their behavior in a mechanically agitated molten aluminum bath. The main goal was to investigate the conditions under which such a master alloy can be fabricated. Additionally, in the third part, the microstructure of the master alloy was characterized by SEM/EDX analysis.

### Theoretical Consideration

In this section, the main idea of this study was presented. As explained above, the addition of some elements, for example, Mn or Cr, remains the main technique for modifying Al-Fe-based intermetallic compounds in aluminum alloys. However, since the formation of new modified compounds occurs from a molten alloy where all elements are distributed more or less uniformly, only a small part of these elements contributes to the modifying effect, while most remain in solution. Thus, the mechanism of such a modification can be described in the following way: In the first stage, as the temperature is decreasing, atoms of aluminum and modifying elements form clusters in the melt followed by the nuclei of high-temperature compounds when the clusters reach a critical size. Then, on a further temperature decrease, atoms of Si and/or Fe start to bond to the nuclei to form particles of four or more component-based compounds which grow as the temperature further decreases. As the formation of clusters occurs, nuclei and their growth occur under diffusion control, this is one of the reasons why the chemical composition and morphology of intermetallic compounds depend on the diffusion coefficients of elements and the time available until full solidification. Although information on the diffusion coefficients of elements in molten aluminum are very limited, the available data indicate that diffusion coefficients differ significantly from one element to another and even influence each other [11]. Obviously, under the above conditions, controlling the morphology of intermetallic compounds is very difficult, if not impossible.

The main idea proposed in the present paper was to add CrSi_2_ microparticles to the molten aluminum alloy, which would, as expected, serve as already existing nuclei for intermetallic compounds during melt solidification. Figure 1 explains why this effect can be achieved. After addition of CrSi_2_ particles into the aluminum melt, CrSi_2_ started to react with the molten aluminum accompanied by the dissolution of Al in CrSi_2_ particles and the formation of new intermetallic compounds. The Al-Cr-Si ternary phase diagram and the results of the earlier investigations provide important information on this issue. Gupta [12] found that, at high temperatures, the CrSi_2_ intermetallic phase has a high solubility of up to 26 at.% of Al, which replaces Si in the CrSi_2_ lattice. As the particle becomes enriched with Al and Cr, a two-component Al-Cr compound, such as Al_45_Cr_7_ and Al_4_Cr, can be formed, particularly on the particle surface. Once formed, such a compound layer may have a protecting effect against the further dissolution of CrSi_2_ particles, as show in Figure 1a. Examining the possibility of fabricating CrSi_2_-particle-containing aluminum master alloys was one of the goals of the present study.

When such a master alloy is added to a molten alloy containing Fe and/or Si, an Al-Cr shell and a CrSi_2_ core, it can dissolve in the aluminum melt, creating areas enriched with Si and Cr, as indicated as Zone S in Figure 1b. For example, molten aluminum (L) can react with CrSi_2_ according to the following reaction [12]:(1)$$L+{\mathrm{CrSi}}_{2}\;⇆{\;\alpha }_{\mathrm{Al}}+\left(\mathrm{Si}\right)\;$$

In the equilibrium state, the chemical composition of the α_Al_ phase is dependent on temperature. For example, at 800 °C the α phase is composed of Cr 24.5, Al 60.9 and Si 14.5 (all in at.%) [13]. Thus, after adding such a master alloy to the Fe-containing aluminum melt, gradients in iron chromium and silicon concentration must exist in the vicinity of the particles. In the presence of the Fe atoms in the melt, as the melt temperature decreases during solidification, the nuclei of Al-Cr-Fe-Si compounds are expected to be formed around the CrSi_2_ particles due to the higher concentrations of Cr and Si here. Thus, the CrSi_2_ particles can promote the nucleation of Cr-modified Al-Fe-Si compounds at much smaller concentrations of Cr than those which are needed to obtain a similar effect when Cr is entirely dissolved in the master alloy.

## 2. Experimental Setup and Procedure

### 2.1. Synthesis of CrSi_2_ Particles with Ball Milling Treatment

Planetary ball milling equipment (P-6; Fritch Co., Ltd., Tokyo, Japan) with a tungsten carbide vial was used to synthesize CrSi_2_ powder particles. To do this, a mixture of silicon and chromium powder at an atomic ratio of 2:1 and 50 milling balls made of tungsten carbide with a diameter of 10 mm were put into the vial. Figure 2 shows typical SEM views of the Cr and Si particles used in the present study. The ball milling conditions were as follows: First, the vial was rotated at a speed of 400 rpm for 10 min, followed by a 10-min pause. Then, the vial was rotated for 10 min again but in the opposite direction. The pause was needed to avoid overheating of the sample during the milling operation. This cycle was repeated 10, 20 or 30 times. Hence, the total time of treatment including pauses was 5, 10 or 15 h to synthesize 40~50 g of CrSi_2_ powder. As-synthesized powders were characterized for phase composition and particle size distribution using X-ray Diffractometer Smart Lab 9 kW (Rigaku, Tokyo, Japan) and particle size analyzer (Microtrac, MT3300, Osaka, Japan), respectively. For the sake of comparison, a commercially available powder of CrSi_2_ was also used for characterization and the following master alloy fabrication. The CrSi_2_ powder was purchased from FIJIFILM Wako Pure Chemical Co. (Sendai, Japan) In the following, the purchased and synthesized particles were referred to as “powder A” and “powder B”, respectively.

### 2.2. Incorporation of CrSi_2_ Particles into Aluminum Melt

Figure 3 is a schematic drawing of experimental setup used to fabricate CrSi_2_ particle-containing master alloy. The setup comprises two main parts: an electrical furnace and a rotary stirrer. About of 1.5 kg of aluminum (99.9%) was melted in a CP crucible (I.D. 98 mm and height 200 mm) placed inside the furnace. Aluminum contained Fe, Si and Cu impurities in amounts of 0.06, 0.03 and 0.002 wt.%, respectively. After reaching a target temperature varied in a range of 750~900 °C, a Nb-made two-blade impeller was immersed in the molten bath to stir it with a rotation speed of 500 rpm. Under these conditions, a deep vortex was formed on the free surface of molten bath around the impeller shaft (Diam. 8 mm). The blade width and height were 48 and 20 mm, respectively. A predetermined amount of CrSi_2_ particles of powder A or B was added onto the surface of vortex, and then the molten bath was stirred for a certain period of time, followed by casting into an iron mold to fabricate an ingot. The stirring time was set to 7, 12, 20 and 30 min. Finally, samples were cut out of the fabricated master alloy ingot and analyzed for chemical composition and microstructure by using an optical emission spectrometer (Carl Zeiss Co., Ltd., Tokyo, Japan) and SEM/EDX instruments (JEOL Ltd., Tokyov, Japan).

## 3. Results

### 3.1. CrSi_2_ Particle Synthesis

The results of XRD analysis revealed that 15 h of ball milling is enough to synthesize the CrSi_2_ particles from the original Si and Cr powders. Figure 4a,b shows the XRD patterns of powder samples after 5 and 15 h of ball milling. For comparison, an XRD pattern of the commercial CrSi_2_ powder is presented in Figure 4c. As can be seen from these figures, although two CrSi_2_-related peaks were clearly seen in the XRD pattern after 5 h of milling, a significant amount of Cr was still detectable in the powder samples. As the milling time became longer, the intensity of the Cr peak was found to decrease, and after 15 h of milling, no Cr was detected in the samples. It is worth noting that in our ball milling process, 20 and 25 h of milling were also used. Even after such a long treatment, Si peaks were still detected. However, silicon is not an impurity because the CrSi_2_-containing mater alloy was intended to be used for addition to Al-Si alloys. Instead, a number of peaks associated with the CrSi_2_ compound are readily observed in Figure 4. These peaks are rather broadened compared to those of the purchased CrSi_2_ powder. Peak broadening is very typical for materials synthesized by ball milling. This is associated with the lattice strain and the nanocrystalline structure of these materials. Measurements revealed that the CrSi_2_ particles of powder A have a narrower distribution in size compared to those of powder B. In the former case, the distribution was bimodal with two peaks at 5 and 20 μm, as shown in Figure 5a. On the other hand, powder B had a broader multimodal distribution with the maximum probability at 40 μm. Part of the particles was in the submicron range, as seen in Figure 5b. This was also clear from Figure 6 showing the typical SEM views of the CrSi_2_ particles of powders A (a) and B (b), respectively. In the latter case, many fine submicron particles were observed in the SEM image. These fine particles form agglomerates the size of which can be as large as 100 μm.

### 3.2. Incorporation of Particles in Melt

Next, the incorporation efficiency of each powder by aluminum melt was investigated. The incorporation efficiency was defined as a weight fraction of CrSi_2_ powder transferred to the melt during impeller agitation. To determine the weight fraction, α, the amount of silicon in the ingot after each experiment was measured, and then the obtained value was divided by the amount of silicon added to the melt with CrSi_2_ powder according to Equation (2).
(2)$$\alpha =\frac{W_{\mathrm{in}}\left(\left[\mathrm{Si}\right]-{\left[\mathrm{Si}\right]}_{0}\right)}{100{\;W}_{\mathrm{Si},p}}\;$$
where W_in_ is the weight of the ingot, W_Si,p_ is the weight of silicon added to the melt with CrSi_2_ powder, [Si] and [Si]_0_ are the wt.% of Si in the ingot after and before experiments.

Figure 7 presents the variation of α over time for powder B at the melt temperature of 800 °C. It was seen that the incorporation efficiency increased up to 0.67 during the first 20 min of treatment and then remained almost constant. Three reasons were assumed to be responsible for this relationship. The first one was that fine CrSi_2_ particles cannot enter the melt because of the interfacial resistance. Second, some CrSi_2_ particles can sinter after adding onto the vortex surface. Obviously, the long time during which the particles stay close together in the vortex and in high temperatures should promote strong interactions between them that can result in their sintering. Because the density of CrSi_2_ is much higher than that of aluminum, such sintered lumps can enter the melt and sink down, being neither dispersed nor dissolved in it. The third reason was that part of the particles sticks to the impeller shaft during the treatment. Based on the results of Figure 7, the time of the stirring treatment was set to 20 min in the following experiments.

Figure 8 shows the effect of temperature on the particle incorporation efficiency for both powders A and B. The data revealed that the efficiency increased with the melt temperature, reaching 0.9 at 900 °C when the particles of powder A were added to the melt. It is notable that the incorporation efficiency of powder B was less than that of powder A. This difference varied with the melt temperature being the smallest at 900 °C and the largest at 850 °C. This was most likely because powder B comprised much finer particles than powder A. It is well known that the finer the particles, the more difficult they are to incorporate into molten metals due to a very high interfacial resistance. However, as the temperature increases, the reaction rate between molten aluminum and CrSi_2_ particles becomes faster. This can result in the formation of an intermediate layer at the interface between the aluminum melt and CrSi_2_ particles that should improve its wettability and incorporation efficiency.

### 3.3. Master Alloy Microstructure

The microstructure of master alloys provides insight into the behavior of CrSi_2_ particles in the melt after their incorporation. Typical data of microstructural observations are presented in the figures below, and it should be mentioned that both the particles of A and B types were used, but, following Figure 9, Figure 10, Figure 11, Figure 12, Figure 13 and Figure 14 and Table 1 and Table 2, the present results were obtained with the synthesized particles of the B type. Generally, at least three types of intermetallic compounds were observed in the samples of master alloys. The first type had relatively small particles, and the morphology and chemical composition of which were found to be dependent on the temperature and agitation time of the melt during the incorporation treatment. Figure 9 presents typical SEM views of particles of this type. They are observed as white particles in this figure. The particle shape is seen to be changed from near-circular to strongly elongated as the melt temperature increases. The corresponding average chemical composition and the aspect ratio of the particles are summarized in Table 1 and Table 2 for different temperatures and agitation times. The aspect ratio is defined as the length of a particle divided by its maximum width. It was clearly seen that the average aspect ratio increased with the agitation time, but no clear dependence on the melt temperature was found. Another interesting finding was that the concentration of Cr in the intermetallic compounds became higher with the increase in the melt temperature and the agitation time, while that of iron, on the contrary, became lower as the melt temperature and the agitation time increased. This is readily seen from a comparison of the data in Table 1 and Table 2. In these tables, the third column shows the atomic concentration of Si in the intermetallic compounds. Clearly, this concentration was much higher than it would have been if the compounds were CrSi_2_ particles. This suggested that these fine particles have been formed as a result of the dissolution of CrSi_2_ in the aluminum melt, followed by the formation of new Al-Si-Fe-Cr compounds during the melt solidification.

The second type of particles is relatively large intermetallic compounds composed of the primary CrSi_2_ particles and Al-Cr intermetallic compounds. Figure 10 is a typical SEM image of such particles. Gray-colored particles of irregular shapes are Al-Cr compounds. It was seen that these particles can be as large as 100 μm, and their size was found to increase with temperature and the agitation time. It was obvious that these particles could only grow up to be such a big size when the original CrSi_2_ particles formed aggregates in the melt. XRD analysis revealed that these large particles were composed mainly of the Al_45_Cr_7_ phase. A typical XRD pattern is shown in Figure 11. Additionally, some small peaks indicated the presence of chromium and silicon in the particles of the second type, as seen in Figure 11. This was also confirmed by SEM/ESX analysis. Figure 12 shows the distribution of Al, Cr and Si in a relatively small particle of the second type. A bright green color indicated areas with a high concentration of Si coexisting with Cr and Al. These areas were most likely to correspond to the remains of the initial CrSi_2_ particles in which Al was dissolved. The other areas were depleted of Si, which shows its preferential dissolution in Al according to Equation (2), as mentioned above. The remaining Cr reacted with Al, producing an Al-Cr compound. This part of particle is colored gray in the SEM view of Figure 12. It was seen that some of such initial particles remained embedded into the bulk of Al-Cr compounds, while the others were located at the interface between the compound particle and aluminum matrix. The results of EDX point analysis showed that Al-Cr compounds were composed of 86.8 at.% of Al and 12.4 at.% of Cr. The concentration of Si in this area was negligibly small, only 0.8 at.%. A ratio of Al to Cr suggested that the composition of Al-Cr compounds was close to Al_45_Cr_7_.

Finally, the third type of particles were rounded and very small particles. Figure 13 is a typical SEM view of these particles showing that their size did not exceed a few micrometers. It was assumed that these were fine original CrSi_2_ particles remaining in Al. It is to be noted that such particles were only found at relatively low temperatures of the melt and short agitation time. For example, the structure shown in Figure 13 was observed in samples produced at 750 °C and an agitation time of 7 min.

Further SEM investigations of such particles showed interesting details regarding their chemical composition and formation mechanism. Figure 14 shows SEM images taken in secondary (a) and backscattering (b) modes as well as the elemental mappings (c–f) of a group of particles. The oval around the particles in Figure 14d indicates an area with increased concentration of Si. This suggests that the dissolution of Si in the Al matrix occurred faster than that of other elements. It is notable also that the particles contained iron. The concentration of iron in the original CrSi_2_ particles was extremely low. Therefore, iron could only enter the particles from aluminum which contained about 0.06% of Fe as an impurity. It was assumed that this phenomenon occurred when the melts were cooled down.

## 4. Discussion

The ball milling time had a major effect on the particle size distribution. Comparing with Figure 6a,b, ball milling was an effective way to produce fine submicron CrSi_2_ particles that have smaller sizes than purchased CrSi_2_ particles. In addition, the agglomeration of fine CrSi_2_ particles during 15 h of the ball milling process was assumed to be responsible for the appearance of coarse particles as large as 100 μm. Similar results were obtained by other research groups. For example, Kumar et al. [14] investigated the effect of the ball milling time ranging from 2 to 100 h on particle morphology and found that the degree of powder particle agglomeration increases with milling time when it is longer than 10 h.

Particles size has a great influence on the incorporation efficiency. The results of our previous research revealed that larger particles have a higher incorporation efficiency, which can be explained according to the following two mechanisms [15]: The first one is an entrainment of particles into a liquid bath due to the oscillating motion of the vortex surface. It was shown that particles with larger diameters penetrated through the gas-liquid free surface of the vortex easier than fine particles. The second mechanism is related to capillary phenomena. The important point of this mechanism is that liquid can rise through capillaries formed in a layer of particles lying on the vortex surface. However, this mechanism requires that the particle surface be wetted with the liquid. In other words, the wetting angle should be less than 90°. Therefore, this mechanism can only be realized if CrSi_2_ has reacted with Al, for example, according to Equation (1), to form a well-wetted layer on the surface of CrSi_2_ particles.

As shown in the result part, three typical compounds were formed after adding the CrSi_2_ particles into the Al melt. In this section, the corresponding formation mechanisms were discussed. Firstly, the dissolution mechanism of added CrSi_2_ particles into the Al melt were discussed to clarify the key parameters influencing the dissolution time of CrSi_2_ particles.

As mentioned in the introduction section, CrSi_2_ compounds can dissolve significant amounts of aluminum. Therefore, the particle dissolution into the melt and the mass transfer of Al from the melt into the particle may proceed simultaneously. Obviously, both these phenomena are controlled by diffusion and should be taken into account. However, because the diffusion coefficients in liquids and solids may differ by two or more orders of magnitude, the dissolution of Al in solid CrSi_2_ particles was neglected in the present analysis. Below is a simplified model of the dissolution of a single CrSi_2_ particle. The model was derived under the following assumptions:The particle is spherical in shape;Dissolution is controlled by the mass transfer of chromium in molten aluminum;The concentration of chromium at the solid–liquid interface is equal to its solubility limit in aluminum at a given temperature.
(3)$$-\rho_{CrSi_{2}}\frac{M_{Cr}}{M_{Cr}+2M_{Si}}\frac{dV_{p}}{dt}=\frac{\rho_{Al}A_{p}k}{100}\left(C_{Cr,e}-C_{Cr}\right)\;\;$$
where $\rho_{CrSi_{2}}$ and $\rho_{Al}$ are the density of the CrSi_2_ particle and the aluminum melt, $M_{Cr}$ and $M_{Si}$ are the atomic weight of chromium and silicon, $V_{p}$ and $A_{P}$ are the volume and surface area of the particle, k is the mass transfer coefficient of chromium in molten aluminum, and $C_{Cr,e}$ and $C_{Cr}$ are the concentration of chromium at the solid–liquid interface and in aluminum melt.

From the insertion of the appropriate expressions for $V_{p}\;\left(\frac{4}{3}\pi r^{3}\right)$, $A_{P}\;\left(4\pi r^{2}\right)$ and *k*, one can derive the following equation:(4)$$-\rho_{CrSi_{2}}\frac{M_{Cr}}{M_{Cr}+2M_{Si}}\frac{r\;dr}{dt}=\frac{\rho_{Al}D_{Cr}}{100}\left(C_{Cr,e}-C_{Cr}\right)\;$$

In Equation (4), *r* is the particle radius and $D_{Cr}$ is the diffusion coefficient of chromium in the aluminum melt determined from Equation (5) [16]:(5)$$D_{Cr}=2.17\times 10^{-5}exp\left(-\frac{88,000}{RT}\right)\;$$

Notice that the mass transfer coefficient, *k*, in Equation (4) was calculated from the Ranz and Marshall correlation for a sphere according to the following equations:(6)$$Sh=2.0+0.6Re^{0.5}Sc^{0.33};\;$$
(7)$$Sh=\frac{kd}{D_{Cr}}\;;$$
(8)$$Re=\frac{u_{P}d}{\nu };\;$$
(9)$$Sc=\frac{\nu }{D_{Cr}}\;$$
where *Sh*, *Re* and *Sc* are the Sherwood, Reynolds and Schmidt numbers, respectively; *d* is the particle diameter; *u_p_* is the sedimentation velocity of particles in the melt; and *ν* is the kinematic viscosity of the melt. It can be shown that the second term of this equation is much smaller than 2.0. Below is a calculation example for CrSi_2_ particles that have two typical diameters, 2 μm and 20 μm.

For fine particles, the sedimentation velocity can be determined from the Stokes law as
(10)$$u_{p}=\frac{d^{2}\left(\rho_{CrSi_{2}}-\rho_{Al}\right)g}{18\mu }\;$$
where *g* is the gravity acceleration and *μ* is the dynamic viscosity of the melt. The calculations were performed using the following data: $\rho_{CrSi_{2}}=5000$ kg/m^3^, $\rho_{Al}=2340$ kg/m^3^, and $\mu =1.18\times 10^{-3}$ Pa ·s. The calculation gave *u_p_* = 4.9 × 10^−6^ m/s and 4.9 × 10^−4^ m/s for particle diameters of 2 μm and 20 μm, respectively. The insertion of these values into Equation (8) gave *Re* = 1.95 × 10^−5^ and 1.95 × 10^−2^ for the particles of 2 μm and 20 μm, respectively. The Schmidt number was calculated to be 94.3 for the above physical properties. Then, one could calculate the second term of the right part of Equation (6), which describes the convective mass transfer. The results gave 0.012 and 0.38 for the particle diameters of 2 and 20 μm, respectively. Therefore, in the case of fine particles with the size of a few microns, the convective mass transfer was negligibly small. For the particles of a few tens of microns in diameter, although the contribution of the convective mass transfer became larger, it was still significantly smaller than that of diffusion and was ignored in the derivation of Equation (4). After integration of Equation (4), one arrived at an Equation (11), describing the time, *t_d_*, required for the complete dissolution of CrSi_2_ particles in the aluminum melt.
(11)$$t_{d}=\frac{100d^{2}M_{Cr}\rho_{CrSi_{2}}}{4\left(M_{Cr}+2M_{Si}\right)\rho_{Al}D_{Cr}C_{Cr,e}}\;$$

Notice that, since *C_Cr_* in Equation (4) is much smaller than *C_Cr,e_*, the former was ignored in deriving Equation (11). The calculation by this equation revealed that the dissolution time was very short. For example, the dissolution time of a 20 μm particle was about 11 s at the melt temperature of 750 °C.

However, as mentioned above, this model did not take into account the possible formation of a solid layer on the original CrSi_2_ particles, which could significantly impede the dissolution rate. For example, as seen in Figure 14, a group of individual particles was surrounded by an area with a higher concentration of Si. This suggested that silicon was transferred from the particles to the melt, most likely due to the reaction (2). As a result, the surface layer of particles became enriched with Al and depleted of Si. This can result in the formation of an Al-Cr compound on the particle surface. This is clearly seen in Figure 10 and Figure 12. Careful observation of these SEM images reveals that such large compounds were probably formed because the CrSi_2_ particles initially formed agglomerates. These particles were observable in these figures as small white spots. Figure 12 reveals that some of these CrSi_2_ particles were embedded into the Al-Cr matrix. Obviously, the Al-Cr compounds are formed when the melt temperature decreases. Therefore, a slight reduction in the melt temperature during the penetration of CrSi_2_ particles in the melt is a desirable condition to enhance the formation of the Al-Cr layer on the particle surface and, thus, to suppress particle dissolution. Another problem to be solved is the suppression of the agglomeration of particles. One possible solution for this is to apply ultrasound vibrations to the melt containing the particle agglomerates. These issues will be the subject of our future investigation.

One more important point to be mentioned was the detection of Fe inside the small particles shown in Figure 12. Although these particles were assumed to be original CrSi_2_ particles, they may contain significant amounts of Al dissolving into the particles according to the mechanism discussed above. After completing the melt stirring, it was poured into a steel mold. During the cooling down process, some amount of Fe, which exists in the melt as an impurity, could dissolve into the particles to eventually produce nuclei of Al-Cr-Fe-Si compounds. Although the system as a whole may be far from equilibrium, an appropriate phase diagram could be helpful in understanding in which direction the system must change to reach equilibrium locally. Figure 15 presents a phase diagram of quaternary Al-Si-Cr-Fe alloys in coordinate temperatures—the chromium concentration was predicted by Thermocalc. The concentrations of silicon and iron were fixed to be 1.5% and 0.1%, respectively. The concentration of chromium was ranged from 0 to 2.5%, and the last value corresponds approximately to the solubility level of Cr in molten aluminum at 800 °C.

When the melt temperature during mechanical stirring was high and its duration was long, CrSi_2_ particles had time to dissolve in the melt, and then, when the melt was cooled down, Al_9_Fe_2_Si_2_ and Al_13_Cr_4_Si_4_ compounds coexisted with liquid and solid aluminum within a narrow temperature range, as shown in Figure 15 by the hatched area. As the solidification of liquid aluminum and the formation of these compounds proceeds simultaneously, they were pushed to the aluminum grain boundaries, as can be seen from Figure 9. On the other hand, when the melt temperature was comparatively low and the duration of mechanical stirring was relatively short, some of the particles could remain in their original form or transformed partly into Al-Cr compounds. In this case, the concentration of Cr near such particles was high and, as seen in Figure 15, iron could only enter the particles through a solid-state transformation. This is why we observed the presence of iron in CrSi_2_ particles in small amounts, as shown in Figure 14.

## 5. Conclusions

In the present work, the fabrication process of master alloys by adding CrSi_2_ particles into molten aluminum was investigated. The master alloys were designed to use it for the modification of Fe-containing intermetallic compounds in aluminum alloys with a higher concentration of iron. CrSi_2_ particles were synthesized by ball milling and then added onto the surface of a vortex formed during the mechanical agitation of an aluminum melt. Factors influencing the particle synthesis performance, the incorporation efficiency into the melt and the morphology of intermetallic compounds in the fabricated master alloys were considered, with the main emphasis being placed on dissolution mechanisms when CrSi_2_ particles entered the Al melt. Based on the above results and discussion, the following conclusions can be drawn from this study:

1. Ball milling is an effective way to produce CrSi_2_ particles, and 15 h of ball milling is enough for original Cr and Si powders to be completely turned into CrSi_2_. When compared to commercial CrSi_2_ powder, the self-synthesized one has a broader size distribution of particles, with the largest ones, up to 100 μm, formed because of agglomeration.

2. CrSi_2_ particle size has a great influence on the incorporation efficiency. Specifically, larger particles penetrate the melt better. This is assumed to be due to a combined effect of capillary and inertial forces acting on particles adhered to the melt surface in the vortex.

3. The low temperature of the melt and the shorter time of mechanical stirring are desirable for the master alloy fabrication because of the suppression of the particle dissolution under these conditions. Nevertheless, the results of theoretical consideration revealed that, even at lower temperatures, the CrSi_2_ particles rapidly dissolve in the melt. This suggested that the fast formation of a layer of Al-Cr compounds on the surface of CrSi_2_ particles is of prime importance for the master alloy fabrication.

4. There are at least three types of intermetallic compounds that were found to exist in the Al melt after addition of CrSi_2_ particles: (1) inclusions of eutectic origin formed at the last stage of crystallization, (2) mixtures of Al-Cr compounds and original CrSi_2_ particles and (3) original CrSi_2_ particles. Furthermore, low melt temperatures and short treatment times were found to favor the fabrication of master alloys because they prevent the dissolution of CrSi_2_ particles into Al melts, which allows one to fabricate the master alloy containing the particles of the second and third types.

5. The above results suggested that a slight reduction in the melt temperature and the introduction of ultrasound vibrations in the melt after addition of CrSi_2_ particles would be favorable to accelerate the formation of the Al-Cr layer on the particle surface and to break the particle agglomerates. These issues will be the subjects of our future works.

## Figures and Tables

**Figure 1 materials-15-07836-f001:**
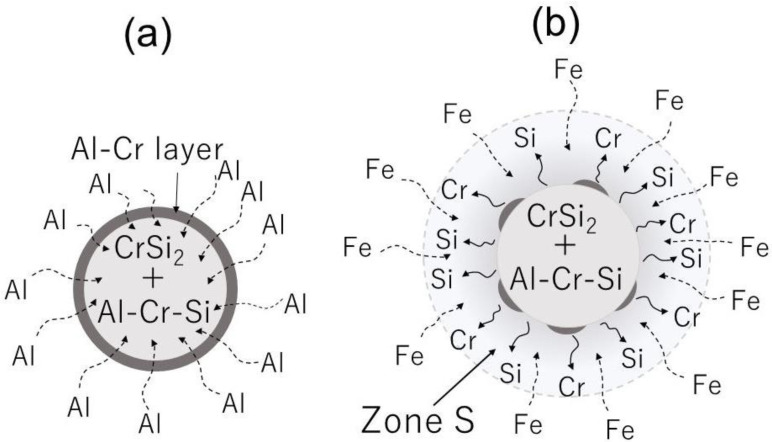
Schematic representation of CrSi_2_ particles in master alloy (**a**) and (**b**) particle dissolution in Al-Fe-Si melt.

**Figure 2 materials-15-07836-f002:**
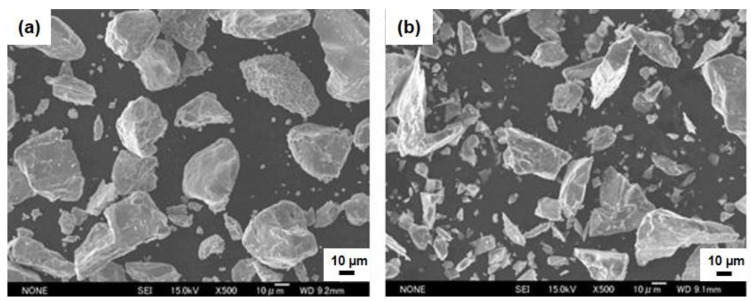
SEM images of original Cr (**a**) and Si (**b**) particles.

**Figure 3 materials-15-07836-f003:**
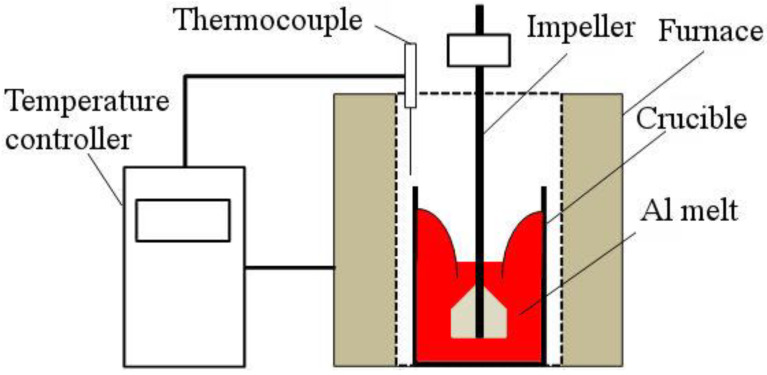
A schematic representation of experimental setup for particle incorporation.

**Figure 4 materials-15-07836-f004:**
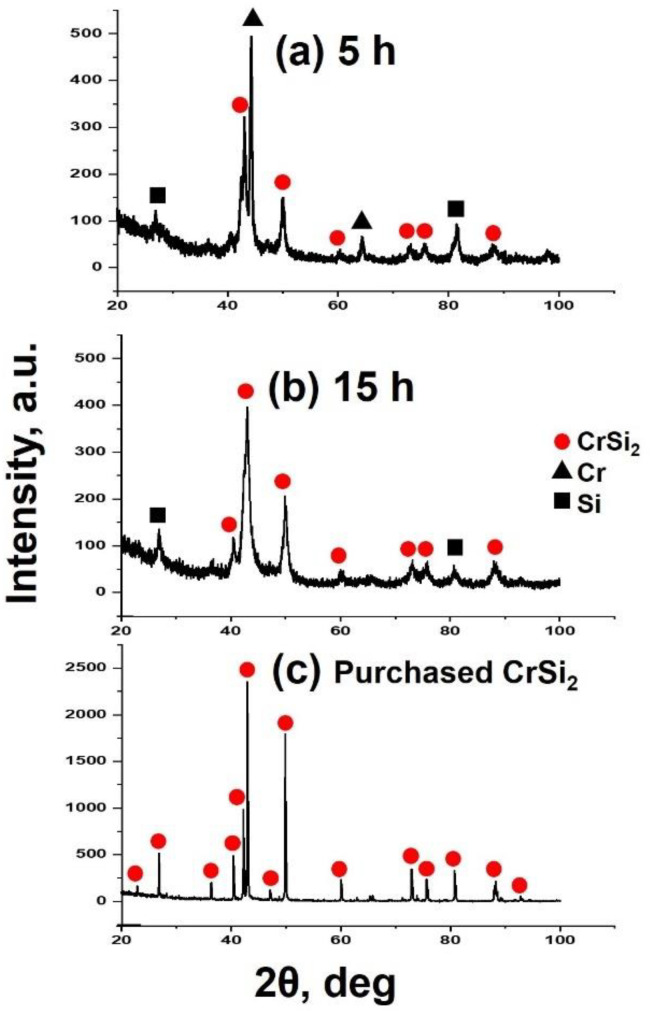
XRD patterns of synthesized particles after different ball milling times: (**a**) 5 h and (**b**) 15 h. XRD pattern of the commercial CrSi_2_ powder: (**c**).

**Figure 5 materials-15-07836-f005:**
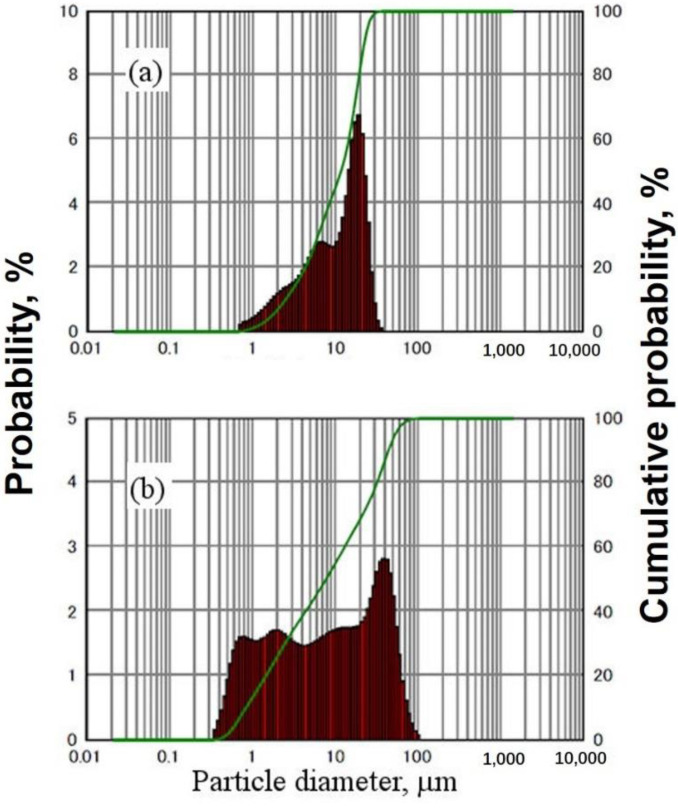
Particle size distribution: (**a**) powder A and (**b**) powder B.

**Figure 6 materials-15-07836-f006:**
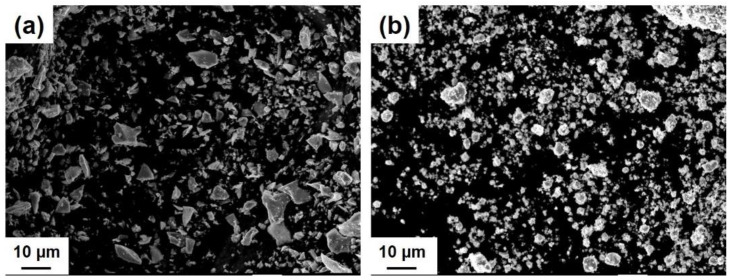
SEM views of particles of powder A (**a**) and B (**b**).

**Figure 7 materials-15-07836-f007:**
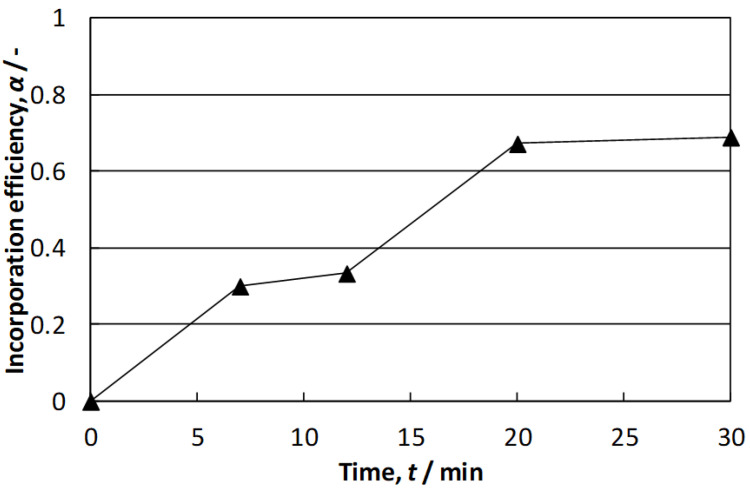
Time variation of incorporation efficiency (Powder B).

**Figure 8 materials-15-07836-f008:**
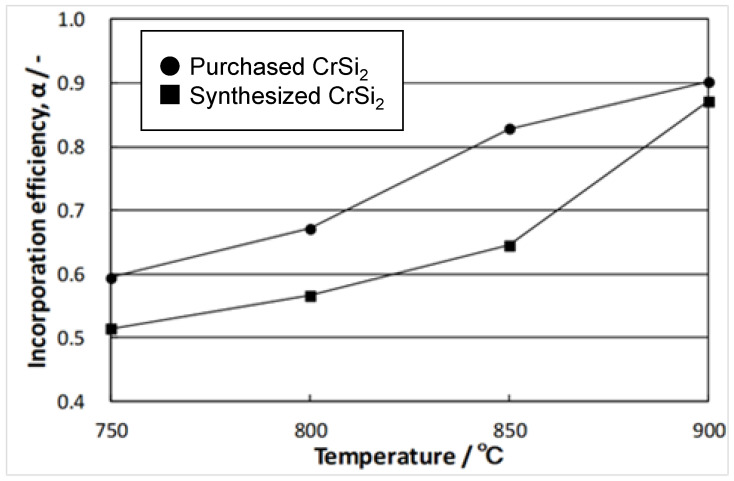
Dependence of incorporation efficiency on melt temperature.

**Figure 9 materials-15-07836-f009:**
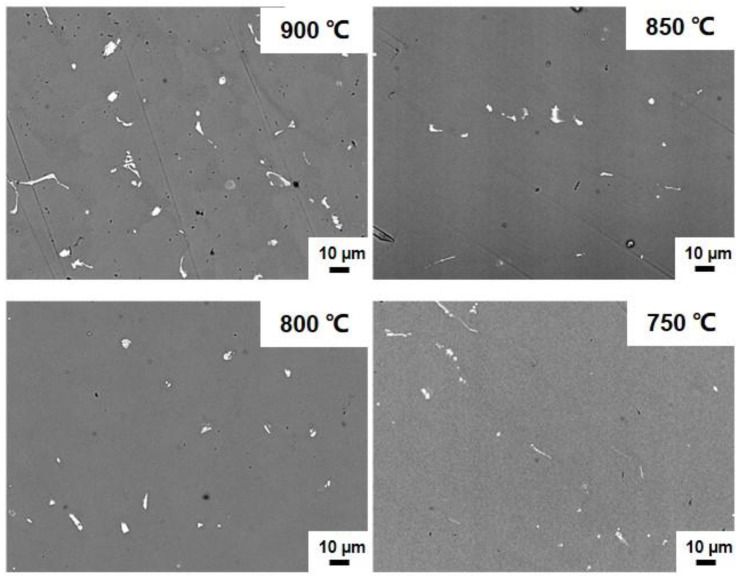
SEM images of microstructure of master alloys produced at various temperatures. Agitation time was 20 min.

**Figure 10 materials-15-07836-f010:**
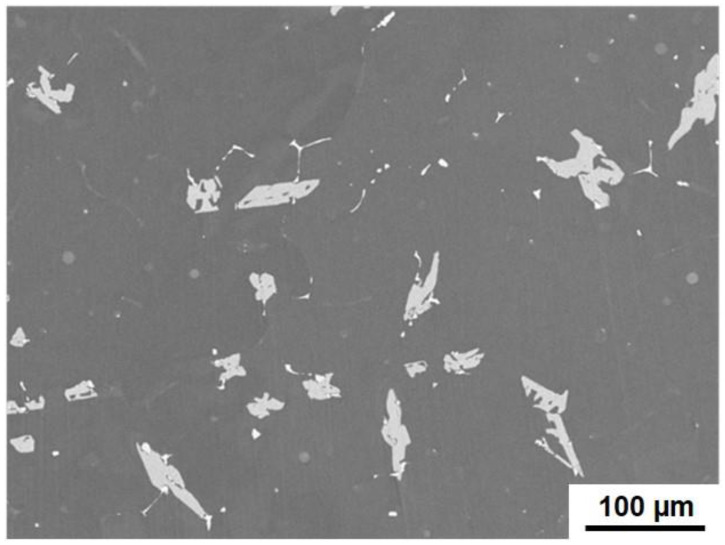
A SEM image of large Al-Cr compounds.

**Figure 11 materials-15-07836-f011:**
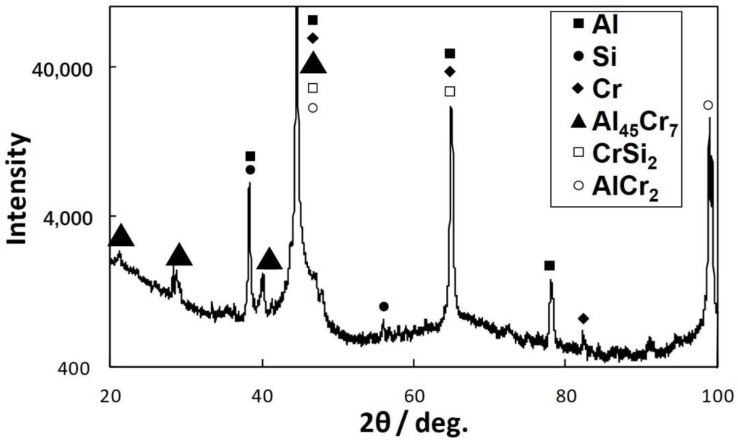
A XRD pattern of sample shown in Figure 9.

**Figure 12 materials-15-07836-f012:**
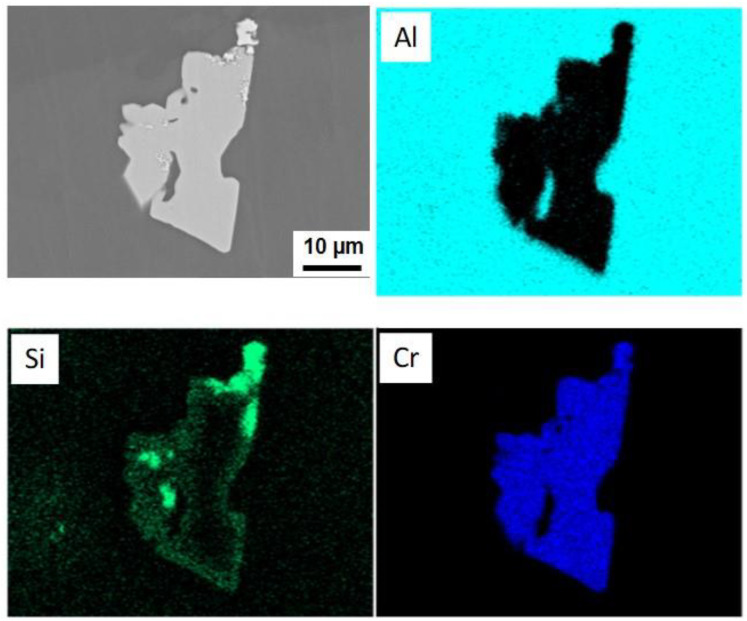
SEM images and element mapping of a particle of the second type.

**Figure 13 materials-15-07836-f013:**
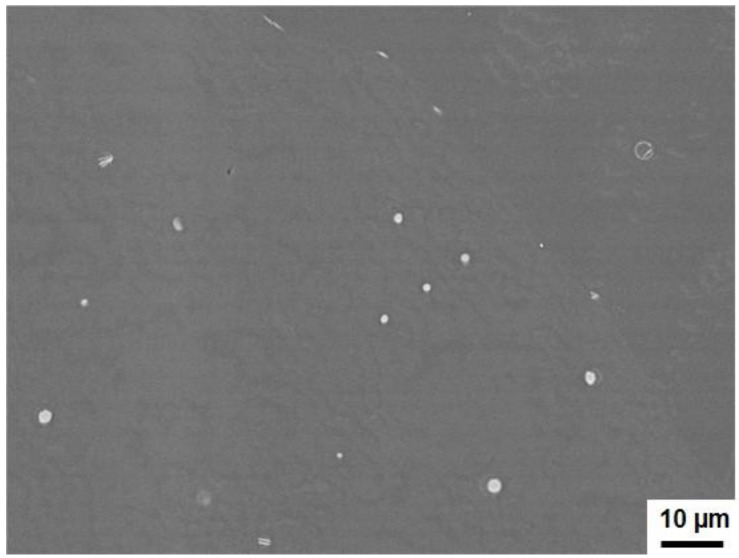
A typical SEM view of particles of third type.

**Figure 14 materials-15-07836-f014:**
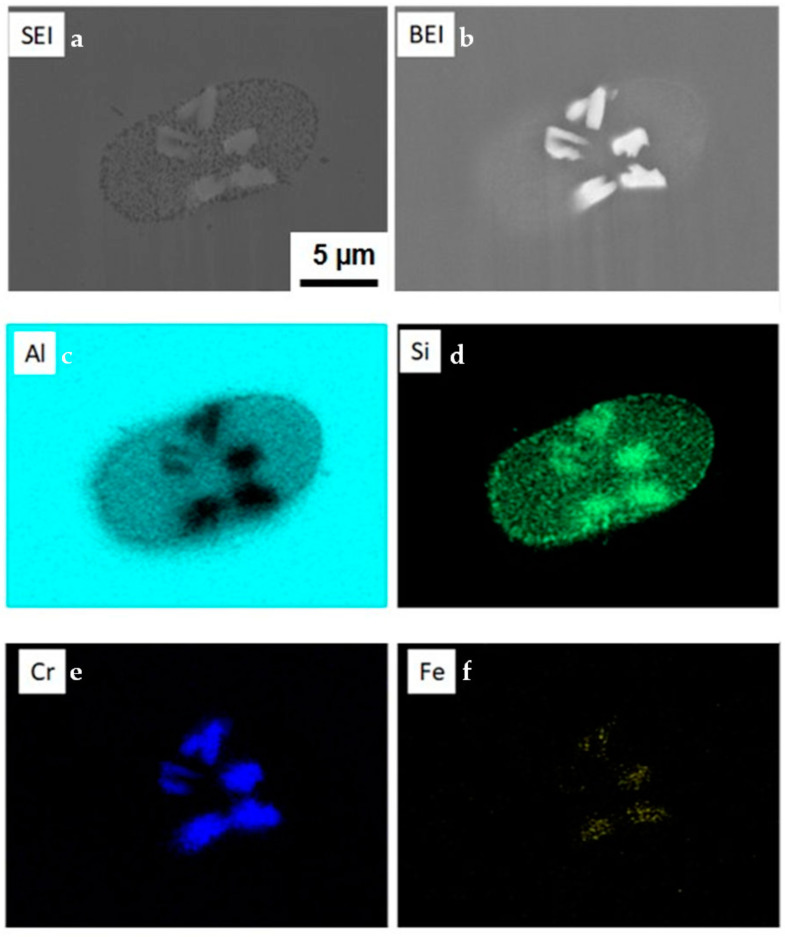
Typical SEM views (**a**,**b**) and elemental mappings (**c**–**f**) of particles of third type.

**Figure 15 materials-15-07836-f015:**
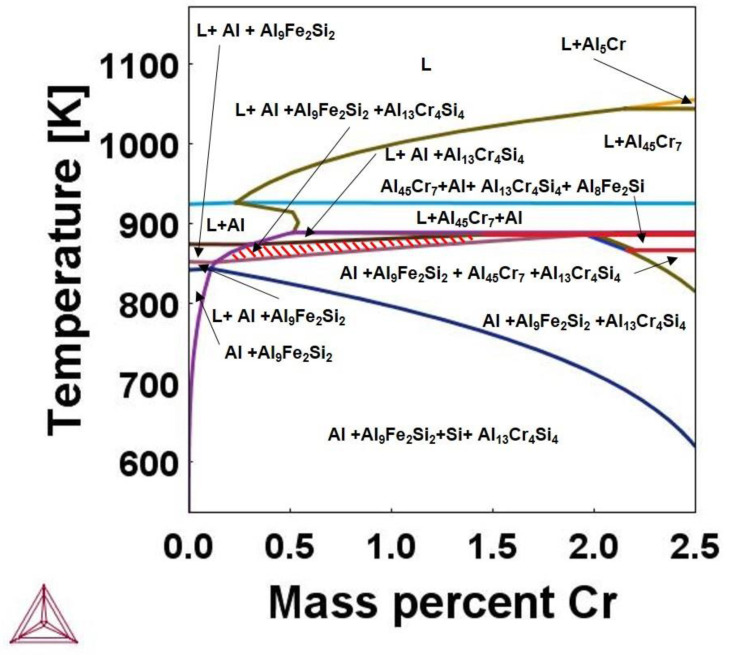
Al-Si-Fe-Cr phase diagram at fixed concentrations of 1.5% of Si and 0.1% of Fe.

**Table 1 materials-15-07836-t001:** Average composition (atm.%) and average aspect ratio of particles of the first type.

Temperature/℃	Al	Si	Cr	Fe	Average Aspect Ratio
900	74.01	11.41	9.7	4.88	3.94
850	75.95	10.31	7.14	6.60	2.33
800	73.27	11.71	8.90	6.11	2.90
750	69.05	16.67	7.24	7.03	2.74

**Table 2 materials-15-07836-t002:** Average composition (atm.%) and average aspect ratio of particles of the first type.

Agitation Time/min.	Al	Si	Cr	Fe	Average Aspect Ratio
7	77.50	12.75	1.45	8.30	1.28
12	78.81	10.42	1.76	7.92	1.68
20	72.84	12.07	8.98	6.09	2.90
30	75.78	10.98	8.49	4.74	3.99

## Data Availability

Not applicable.

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
