# Peer review of "Fabrication of CrSi_2_-Containing Master Alloys for Modification of Fe-Containing Intermetallic Compounds in Aluminum Alloys"

_materials, 2022, doi:10.3390/ma15217836_

Round 1
Reviewer 1 Report
The presented article is of scientific and technical interest. Concerns the problem of aluminum processing. I think that the article should be finalized in accordance with the following recommendations:
1. It is necessary to check the numbering of formulas. The formula number (2) is repeated twice.
2. In the caption of Figure 1, a link to the right figure (b) is missing.
3. The punctuation mark should be placed immediately after the formula, and not after the number of the formula (formulas 6-8).
4. There is no complete inscription in Fig. 16.
5. There is no reference №10 in the text.
6. Reference to the figure must be given before the figure. Must be checked (Figures 2, 9).
7. Check the correctness of the design of captions and table captions in accordance with the requirements for the design of the magazine.
8. Line 347, check the correctness of the reference to Fig. 14. Do need a comma between the word Fig. and the figure number.
9. Figure 1 shows Zone C, which is not mentioned anywhere in the article. Is it worth it then to indicate in the figure?
10. For better clarity and understanding of the processes of the experiment, in addition to the scheme (Fig. 3), a photo of the experimental setup should be indicated.
11. The inscriptions in the figures should be reduced. This is especially true for Fig. 4-6, 16.
12. It would be good to indicate in the article how changes in the composition of the alloy can affect its properties.
Reviewer 2 Report
This is a nice paper, I have a few minor suggestions and corrections:
- it might be worth checking the English in the paper, there are a few issues in places
- line 13, there is a stray "e"
- line 169, do you mean Al or L, if L you should define it in the text
- can the scale bars on the SEM images be more visible, Fig 2b, doesn't seem to have one
- Fig 5 could be added to Fig 4 to help the reader see the different better
- Which measurements were used to determine the particle size (line 235)
- Eq (2) alpha was used in the previous equation, be careful on using symbols twice
- errors on Table 1 and 2
- line 367 wasn't it 15 hrs not 16 ?
line 408, might want to add in what these symbol equal, if you have added them to the equations
- Figure 16 caption is missing an element at the end!
Reviewer 3 Report
The authors have investigated adding CrSi2 to solid melt to modify Fe containing alluminium alloys. The work is well designed and executed, but some issues need to be addresses.
1. The authors investigate addition of both commercial and synthesized powder of CrSi2, namely Powder A is the commercial powder and Powder B is the synthesized powder. They obtained Powder B by milling for 15 hours. However Powder B is not pure, it contains Si and possibly Cr impurity (though Cr is not marked in the XRD). Did the authors try extending the milling time for longer than 15 hours to obtain pure CrSi2?
What was the crystallite size of Powders A and B, maybe the authors could use the Sherrer equation to evaluate this from XRD data?
Also, how did the authors evaluate particle size. The magnification is very low?
2. The authors also need to discuss the influence of the impurities in powder B on the melt microstructure and particle shap.
3. Magnification of the SEM images of the two powders is very low, a higher magnification would enable a proper comparison. Also the magnification needs to be the same, for powder A it is 500, for B it is 1000, maybe 5000 would be better to see the particle shape and agglomeration better. Did the authors perform EDS of the powders to show where the Si impurity is located also?
4. Section 3.3 Microstructure of master alloy.
It is not clear for which powder this analysis is conducted, A, B or both. This needs to be described in detail, as the powder composition and morphology has a significant influence on the properties.
Reviewer 4 Report
In this manuscript authors show fabrication of master alloy for modification of Fe-containing intermetallic compounds in aluminum alloys. The information shown in the manuscript could be interesting and have application potential. The characterization methods are well chosen. The introduction part is well written. However, some points need to be improved:
1. Fig 4 and 5 – please start the graph from 20 deg 2theta.
2. It will be better to consolidate figs 4 and 5 into one figure.
3. Authors mention about peak broadening and some nanostructure, however there are no results about it. It will be good to show some analysis of XRD data after milling and refer it to other results and conclusions.
4. Please improve figures: 7, 10, 11, 13, 14 and 15 by removing the original bar and enlarge scale bar.
My recommendation is: major revision
Regards
Round 2
Reviewer 3 Report
The authors have answered all the questions raised in the review and corrected their work to implement most of the suggestions. I recommend this paper for publication in its present form.
Author Response
Thank you for your comment. English language style and spell have been carefully checked.
Reviewer 4 Report
After revision the manuscript could be accept
My recommendation is: accept
Regards
Author Response
Thank you for your comment. English style and spell have been carefully checked.